# Non-invasive classification of macrophage polarisation by 2P-FLIM and machine learning

**Nuno GB Neto[1,2], Sinead A O'Rourke[1,2,3], Mimi Zhang[4], Hannah K Fitzgerald[3], Aisling Dunne[3,5], Michael G Monaghan[1,2,5,6]\***

[1]Department of Mechanical, Manufacturing and Biomedical Engineering, Trinity College Dublin, Dublin, Ireland; [2]Trinity Centre for Biomedical Engineering, Trinity Biomedical Science Institute, Trinity College Dublin, Dublin, Ireland; [3]School of Biochemistry & Immunology and School of Medicine, Trinity Biomedical Science Institute, Trinity College Dublin, Dublin, Ireland; [4]School of Computer Science and Statistics, Trinity College Dublin, Dublin, Ireland; [5]Advanced Materials for BioEngineering Research (AMBER) Centre, Trinity College Dublin and Royal College of Surgeons in Ireland, Dublin, Ireland; [6]CURAM SFI Research Centre for Medical Devices, National University of Ireland, Galway, Ireland

**Abstract** In this study, we utilise fluorescence lifetime imaging of NAD(P)H-based cellular autofluorescence as a non-invasive modality to classify two contrasting states of human macrophages by proxy of their governing metabolic state. Macrophages derived from human blood-circulating monocytes were polarised using established protocols and metabolically challenged using small molecules to validate their responding metabolic actions in extracellular acidification and oxygen consumption. Large field-of-view images of individual polarised macrophages were obtained using fluorescence lifetime imaging microscopy (FLIM). These were challenged in real time with small-molecule perturbations of metabolism during imaging. We uncovered FLIM parameters that are pronounced under the action of carbonyl cyanide-p-trifluoromethoxyphenylhydrazone (FCCP), which strongly stratifies the phenotype of polarised human macrophages; however, this performance is impacted by donor variability when analysing the data at a single-cell level. The stratification and parameters emanating from a full field-of-view and single-cell FLIM approach serve as the basis for machine learning models. Applying a random forests model, we identify three strongly governing FLIM parameters, achieving an area under the receiver operating characteristics curve (ROC-AUC) value of 0.944 and out-of-bag (OBB) error rate of 16.67% when classifying human macrophages in a full field-of-view image. To conclude, 2P-FLIM with the integration of machine learning models is showed to be a powerful technique for analysis of both human macrophage metabolism and polarisation at full FoV and single-cell level.

**\*For correspondence:**
monaghmi@tcd.ie

**Competing interest:** The authors declare that no competing interests exist.

## Editor's evaluation

The authors introduce a machine learning based classifier for M1 and M2 polarised macrophages based on autofluorescence lifetime parameters excited by two-photon excitation in the NAD(P)H emission band following during uncoupling of oxidative phosphorylation. They have identified a promising direction for use of metabolic imaging for macrophage classification.

## Introduction

Two-photon fluorescence lifetime imaging microscopy (2P-FLIM) is a non-destructive modality that can interrogate exogenous and endogenous fluorophores. 2P-FLIM provides high spatial and temporal resolution to image murine and human cell types, 2D and 3D cell cultures, and biopsies in vitro and in vivo (*Skala et al., 2007*, *Okkelman et al., 2019*, *Neto et al., 2020*). 2P-FLIM allows reduced nicotinamide adenine dinucleotide (NAD(P)H) and flavin adenine dinucleotide (FAD⁺) to be studied and quantified using 2P-FLIM, giving insight into cellular metabolism (*Lakowicz et al., 1992*; *Skala et al., 2007*, *Neto et al., 2020*). The average time for a fluorophore to return to the ground state from the excited state while emitting fluorescence is known as the fluorescence lifetime. NADH and NADPH fluorescence properties are identical, and hence NAD(P)H refers to these intracellular pools combined (*Huang et al., 2002*). NAD(P)H is noted as having a long fluorescence lifetime when enzyme-bound and a short fluorescence lifetime when free in the cytoplasm. NAD(P)H fluorescence properties offer extended information into NAD(P)H protein/enzyme binding (*Lakowicz et al., 1992*). 2P-FLIM facilitates the quantification of NAD(P)H fluorescence lifetimes and their respective proportions. In addition, 2P-FLIM can be used as a microscopy-based spatial approach and applied to limited cell numbers, which is beneficial in assessing limited sample numbers or validating metabolism-based therapies (*Peterson et al., 2018*; *Shields et al., 2020*).

Macrophages are essential components of the host immune response and key regulators of homeostatic function. In addition to host defence, macrophages are intimately involved in tissue homeostasis and play a key role in pathologies including heart failure, diabetes, and cancer (*Mosser and Edwards, 2008*). Macrophages adopt specific polarisation states, ranging in a spectrum, to accomplish various functions and mechanisms of action. Here, classically activated (M1) and alternatively activated (M2) macrophages occupy opposite ends (*Gordon, 2003*; *Mosser and Edwards, 2008*). In vitro, M1 macrophage behaviour can be evoked using IFNγ and/or microbial products such as LPS. M1 macrophage phenotype is defined by secretion of a significant amount of pro-inflammatory cytokines: TNFα and IL-1β enhanced endocytosis and ability to kill intracellular pathogens (*Adams, 1989*; *Martinez et al., 2008*).

M2 macrophage behaviour exists more heterogeneously and can be triggered by IL-4 or IL-13, IL1-β, TGF-β, IL-6, phagocytosis of apoptotic cells, or association with a tumour microenvironment, respectively, generating M2a, M2b, M2c, M2f, and tumour-associated macrophages (TAM) (*Mantovani et al., 2002*; *Mantovani et al., 2017*; *Graney et al., 2020*). M2 macrophages exhibit decreased expression of protein membrane markers, such as CD14 and CCR5, and increased fibronectin-1 production. In addition, M2 macrophages are also characterised by the downregulation of pro-inflammatory cytokines (*Wang et al., 1998*; *Gratchev et al., 2001*; *Martinez et al., 2008*). Most often, macrophage behaviour is assessed by endpoint assays using cytokine measurements, gene analysis, and staining of surface markers. However, there is a gathering shift towards non-invasive modalities to speed up this process and obtain spatio-temporal analysis. Two recent examples of this include the use of Raman microscopy to map the lipidomic spatial signature of polarised macrophages (*Feuerer et al., 2021*), and the use of average fluorescent lifetime parameters to discern murine macrophage phenotype (*Alfonso-García et al., 2016*).

We use an imaging-based approach, primarily focusing on metabolic machinery characteristically employed by polarised human macrophages. Macrophage metabolism poses a huge potential in the next generation of therapeutics for inflammatory disease. Human macrophage function and metabolism are inextricably linked (*Van den Bossche et al., 2017*). IFNγ-activated human macrophages (IFNγ-M1) are primed for enhanced inflammatory responses by stabilising HIF1α levels, activation of the JAK-STAT pathway, and increased production of IL-1β, all of which are dependent on enhanced levels of glycolysis (*Wang et al., 2018*). Alternatively activated human anti-inflammatory macrophages, most often observed in vitro through stimulation with IL-4 (IL-4-M2), are defined by an intact tricarboxylic acid cycle (TCA), enhanced OxPhos, increased fatty acid synthesis (FAS), and fatty acid oxidation (FAO) (*O'Neill et al., 2016*, *Van den Bossche et al., 2017*). In a nutshell, IFNγ-M1 macrophages are more active in glycolysis, while IL-4-M2 macrophages are more dependent on oxidative phosphorylation for their energy production. When assessing metabolism and bioenergetics, most methods are based on extracellular flux assays, measurement of cellular oxygen consumption, exogenous staining, radio-labelling nutrients, and gas chromatography-mass spectrometry (GC-MS). Metabolism probing methods require a substantial amount of sample processing yet still pose limitations due to short-lived

oxidative metabolites (*Vivekanandan-Giri et al., 2011*; *Koo et al., 2016*; *Fall et al., 2020*; *Ma et al., 2020*).

2P-FLIM imaging of NAD(P)H acquires five fluorescence lifetime variables ( $\tau_1$, $\tau_2$, $\alpha_1$, $\alpha_2$, $\tau_{avg}$) and one fluorescence intensity-based variable (optical redox ratio [ORR]) descriptive of the polarisation linked-metabolic state of IFNγ-M1 and IL-4-M2 macrophage phenotypes. Furthermore, additional information can be obtained from these measurements by real-time perturbation of basal metabolism and metabolic capacity. To achieve this, sequential knockdown of intercellular pathways and uncomplexing of mitochondrial coupling is performed in this study via metabolic inhibitors.

For data visualisation purposes, we applied the Uniform Manifold Approximate and Projection (UMAP) technique, in favour over principal component analysis (PCA), due to its processing speed, capability to preserve the global and local structures of the data, and ability to use non-metric distance functions (*McInnes et al., 2018*). For in-depth statistical analysis, machine learning algorithms are employed for classification/supervised pattern recognition (*Mohri and Rostamizadeh, 2012*). Machine learning algorithms have been used to characterise DNA- and RNA-binding proteins, determine genetic and epigenetic contributions of antibody repertoire diversity, and classify chronic periodontitis patients based on their immune cell response to ex vivo stimulation with ligands (*Alipanahi et al., 2015*; *Bolland et al., 2016*; *Culos et al., 2020*). In addition, machine learning algorithms can be used to group samples into different classes (e.g., in this study IFNγ-M1 vs. IL-4-M2 based on 2P-FLIM variables) whilst determining which variables are the most important for this task (*Touw et al., 2013*). For our work, we employed the random forests algorithm for the classification task due to its high prediction accuracy, robustness to outliers, and ability to obtain the relative importance of each variable (*Breiman, 2001*; *Verikas et al., 2011*). Another advantage of the random forests model is that we can use the out-of-bag (OOB) error estimate to determine the values of the hyper-parameters, no need to set aside an independent dataset for validation. This makes random forests especially suitable for small datasets, where we do not have additional data for validating the model. The hyper-parameters of the random forests model are the number of trees (*ntree*) and the number of variables selected for the best split at each node (*mtry*). The OOB error of the final optimal random forests model serves as the estimate of the model's prediction error.

The trained random forests model will be used to distinguish between the different populations of macrophages and to measure which 2P-FLIM variables are the most important for this differentiation. In our study, we used all data points obtained from both full field-of-view (FoV) and donor-specific single-cell images for training and validating the random forests model. Thus, there is no extra data available for testing our trained model. Therefore, to demonstrate the predictive ability and efficiency of the trained model, we calculated the area under the receiver operating characteristics curve (ROC-AUC), in which the specificity and sensitivity of the trained model are plotted. The ROC-AUC statistic is a performance evaluation metric for (binary) classification models. ROC curves plot the true positive rate against the false-positive rate at various threshold values. The AUC is the measure of the ability of a classifier to distinguish between two classes and is used as a summary of the ROC curve. An ROC-AUC value less than 0.5 suggests no discrimination, 0.7–0.8 is considered acceptable, 0.8–0.9 is considered excellent, and more than 0.9 is considered outstanding (*Mandrekar, 2010*). Similar applications of machine learning methods have been explored by Walsh et al. for classification of activated t-cells and Qian et al. for quality control of cardiomyocyte differentiation (*Walsh et al., 2020*, *Qian et al., 2021*).

We hypothesise that 2P-FLIM of NAD(P)H and FAD$^+$ provides quantitative information to evaluate and identify human-derived macrophage polarisation by proxy of their metabolism. 2P-FLIM of NAD(P)H and FAD$^+$ has strong clinical potential fuelled by the emergence of metabolic approaches to treat disease and inflammation. To test our hypothesis, we derived human macrophages from blood-circulating monocytes and polarised them into IFNγ-M1 or IL-4-M2 macrophages. We confirmed human macrophage cytokine and gene expression-related polarisation, while human macrophage metabolic behaviour was assessed via traditional extracellular flux assays. Finally, 2P-FLIM was applied during which real-time responses of photonic variables to metabolism-challenging small molecules were measured. This study establishes 2P-FLIM as a method to discriminate IFNγ-M1 and IL-4-M2 macrophages, which is most pronounced by macrophages' differential responses when treated with carbonyl cyanide-p-trifluoromethoxyphenylhydrazone (FCCP). This allows the accurate classification of macrophage polarisation using machine learning methods, for example, the random forests model,

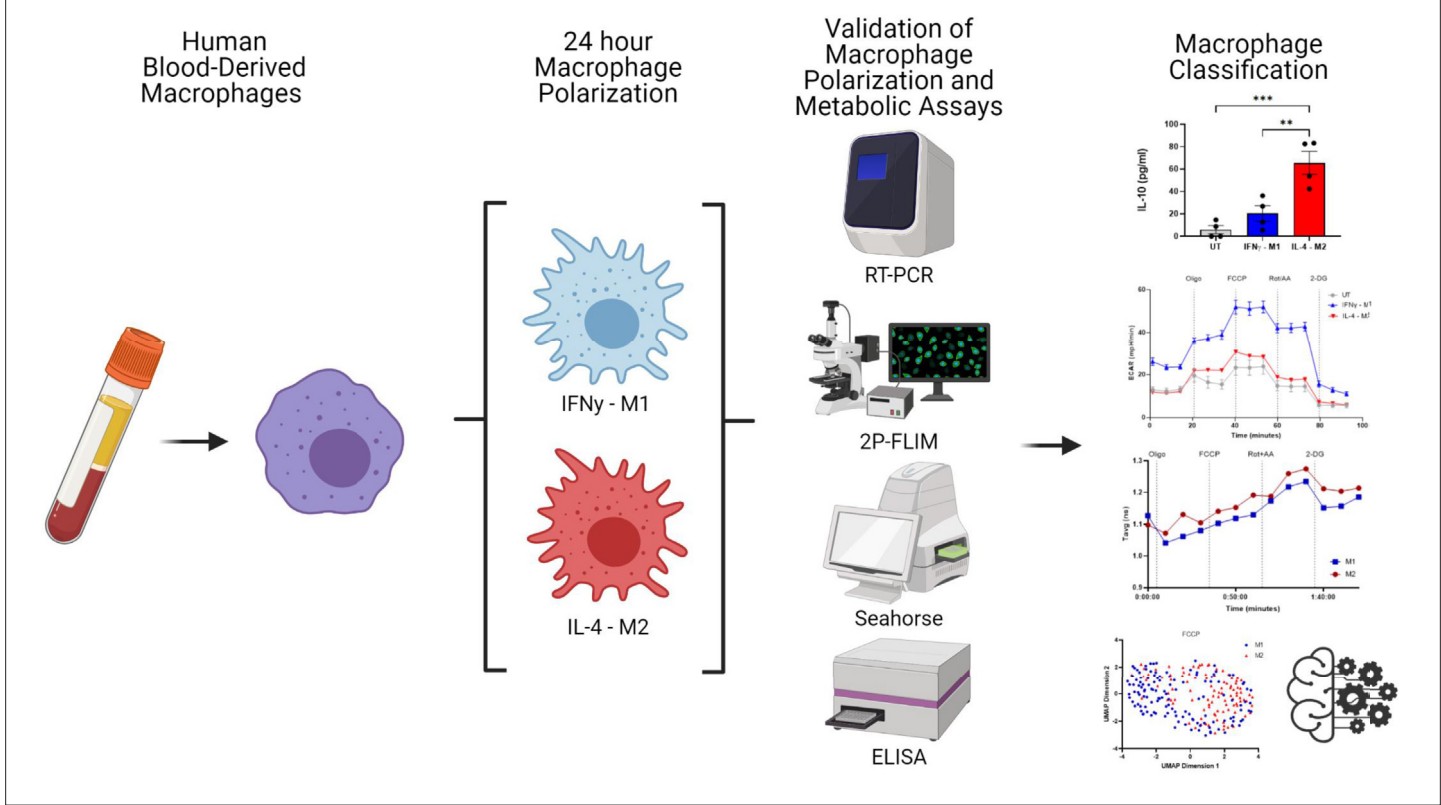

**Figure 1.** Overview of experimental work. Image created using biorender.com.

and to evaluate single-cell heterogeneity. Our work reports, for the first time, the classification of human macrophages by random forests using data obtained from real-time metabolic perturbations triggered during 2P-FLIM.

## Results

### Macrophage polarisation with IFNγ and IL-4 induces metabolic reprograming

Human blood-derived macrophages were polarised by incubating in cell culture media containing IFNγ (M1) or IL-4 (M2) for 24 hr. Polarisation was confirmed using ELISA and RT-PCR. Cellular metabolic activity was analysed using a sequence of metabolic enzyme inhibitors, and the inhibitors' effect was measured by extracellular acidification ratio (ECAR), oxygen consumption ratio (OCR), and 2P-FLIM (*Figure 1*).

A slightly higher amount of TNFα production was obtained for IFNγ-M1 when compared with IL-4-M2 macrophages. Regarding IL-10, a statistically higher production was measured in IL-4-M2 when compared with IFNγ-M1 and untreated macrophages (*Figure 2A and D*). For gene expression, a higher amount of CXCL9 and a statistically significant increase in CXCL10 in IFNγ-M1 macrophages were observed (*Figure 2B and C*). In addition, MRC1 and CCL13 were further expressed in IL-4-M2 macrophages when compared with untreated and IFNγ-M1 macrophages (*Figure 2E and F*). IFNγ-M1 macrophages have a higher dependence on aerobic glycolysis, whilst IL-4-M2 macrophages are more reliant on oxidative phosphorylation. We used ECAR and OCRs to certify this metabolic behaviour, which is linked to macrophage polarisation. For ECAR and OCR, we used four different metabolic modulators in succession, oligomycin, FCCP, rotenone + antimycin A (Rot+AA), and 2-deoxy-D-glucose (2-DG), to evaluate cellular metabolism (*Figure 2*). IFNγ-M1 macrophages exhibited a higher ECAR and lower OCR in response to the treatments added, whilst IL-4-M2 and untreated macrophages had lower ECAR and higher OCR (*Figure 2G and J*). After plotting ECAR and OCR curves, the areas under

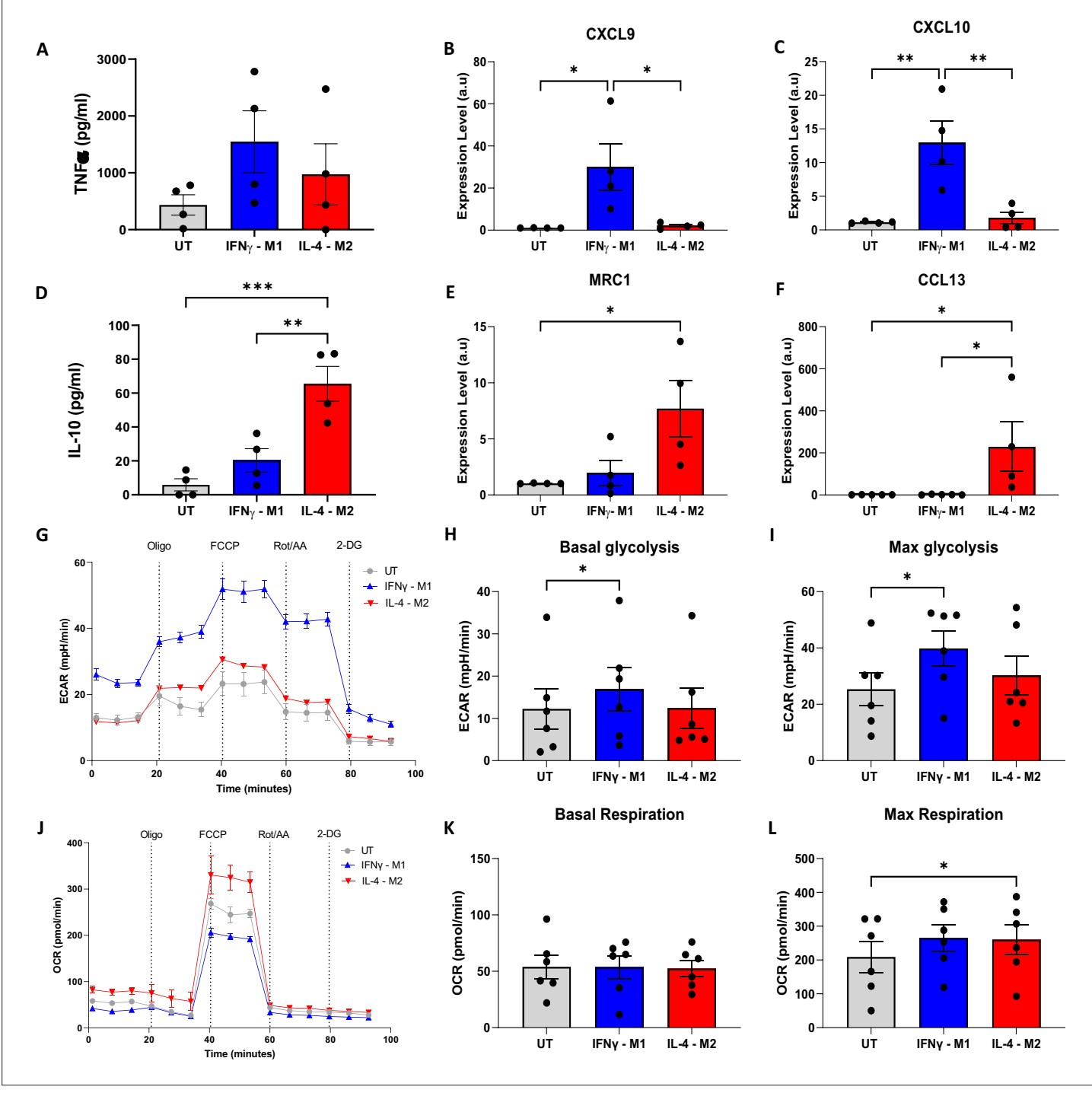

**Figure 2.** Validation of macrophage polarisation and metabolic profiling of IFNγ-M1, IL-4-M2, and untreated (UT) macrophages. (**A, B**) ELISA of inflammatory cytokine TNFα and anti-inflammatory IL-10 in IFNγ-M1, IL-4-M2, and UT macrophages. (**C–F**) Evaluation of CXCL9, MRC1, CXCL10, and CCL13 gene expression in IFNγ-M1, IL-4-M2, and UT macrophages. (**G,J** ) Extracellular acidification ratio (ECAR) and oxygen consumption ratio (OCR) profile of IFNγ-M1, IL-4-M2, and UT macrophages when treated sequentially with oligomycin, carbonyl cyanide-p-trifluoromethoxyphenylhydrazone (FCCP), rotenone + antimycin A, and 2-deoxy-D-glucose (2-DG). (**H, I, K, L**) Area under the curve (AUC) values calculated from ECAR and OCR between each treatment. Data displayed as average ± SD. Statistical significance verified by one-way ANOVA with *p<0.05, **p<0.01, ***p<0.001 to show significance for N = 6 donors.

The online version of this article includes the following source data and figure supplement(s) for figure 2:

**Source data 1.** ELISA, gene expression, extracellular acidification ratio, and oxygen consumption ratio measurements for each replicate data and

*Figure 2 continued on next page*

*Figure 2 continued*

details of statistical tests and chosen parameters.

**Figure supplement 1.** Validation of macrophage polarisation using flow cytometry.

**Figure supplement 1—source data 1.** Flow cytometry surface markers measurement and details of statistical tests and chosen parameters.

the curves (AUC) were measured to reflect basal glycolysis, maximal glycolysis, basal respiration, and maximal respiration.

IFNγ-M1 macrophages have a statistically significant increase of basal and max glycolysis when compared with untreated macrophages (*Figure 2H and I*). In addition, all macrophage types have similar basal respiration, whilst IL-4-M2 macrophages have a statistically significant increase in max respiration when compared with untreated macrophages (*Figure 2K and L*).

## 2P-FLIM captures metabolic shifts on IFNγ and IL-4-treated macrophages

2P-FLIM harvests NAD(P)H and FAD⁺ autofluorescence to infer cellular metabolism. NAD(P)H enzyme-bound state is characterised by a longer fluorescence lifetime, whilst NAD(P)H free-state has a shorter fluorescence lifetime. NAD(P)H and FAD⁺ fluorescence intensities are measured in order to calculate the ORR (*Equation 3*). These fluorescence features enable the distinction between an OxPhos or glycolytic-dependent metabolism (*Skala et al., 1992*, *Okkelman et al., 2019*, *Schaefer et al., 2019*; *Floudas et al., 2020*; *Neto et al., 2020*; *Walsh et al., 2020*, *Perottoni et al., 2021*).

For this experiment, we seeded unpolarised macrophages in ibidi Luer μ-slides in static conditions and polarised the macrophages using IFNγ or IL-4 for 24 hr. These macrophages are derived from the same donors as per those presented in *Figure 2*. Afterwards, we followed the same subjection of metabolic enzymatic inhibitors applied in the ECAR/OCR experiment in which we treated the macrophages with oligomycin, FCCP, Rot + AA, and 2-DG. During the time course of the experiments, the field of view was maintained so as to record single-cell metabolic variations (*Figure 3A*, *Figure 3—figure supplements 1 and 2*).

We derived the average fluorescence lifetime ($\tau_{avg}$) and ORR of IFNγ-M1 and IL-4-M2 macrophages from the full FoV of 2P-FLIM data and observed an increasing trend of $\tau_{avg}$ in response to the application metabolic enzymatic inhibitors. With the exception of 2-DG, in which a decrease of $\tau_{avg}$ was observed for both macrophage phenotypes (*Figure 3B*). Regarding ORR, there is a slight decreasing trend of ORR, followed by a raise in ORR with the 2-DG treatment for IFNγ-M1 macrophages. For IL-4-M2 macrophages, there is a decrease in ORR with the oligomycin treatment, followed by stabilisation with FCCP and Rot + AA, and finally an increase elicited by 2-DG (*Figure 3C*). In addition, we utilised phasor analysis on the raw FLIM data. Here, we plotted the phasor maps while fixing the lifetimes at 3.4 ns and 0.4 ns as indicated in literature (*Ranjit et al., 2018*). Furthermore, we generated histogram plots that showcase the distribution of the data in the phasor plot as well as the difference between IFNγ-M1 and IL-4-M2 macrophages (*Figure 3—figure supplement 3*). We compiled all the full FoV 2P-FLIM variables: $\tau_1$, $\tau_2$, $\alpha_1$, $\alpha_2$, $\tau_{avg}$, and ORR into a representative z-score heatmap, stratified according to macrophage type and metabolic inhibitors across all donors. IFNγ-M1 macrophages have lower $\tau_1$, $\tau_2$, $\tau_{avg}$, and ORR values when compared with IL-4-M2 macrophages (*Figure 3D*).

## 2P-FLIM variables allow the classification of IFNγ-M1 and IL-4-M2 macrophages

UMAP was applied to full FoV 2P-FLIM variables associated with IFNγ-M1 and IL-4-M2 macrophages as a data visualisation tool. The coordinates for each image were defined using a cosine distance function computed using the 2P-FLIM variables: $\tau_{avg}$, $\tau_1$, $\tau_2$, $\alpha_1$, $\alpha_2$, and ORR (*Figure 3E*). UMAP representation of 2P-FLIM variables acquired during FCCP treatment provides a separation between IFNγ-M1 and IL-4-M2 macrophages (*Figure 3F*). This segregation is also observed when applying PCA analysis on FCCP treated human macrophages. However, this separation is not observed for t-SNE analysis (*Figure 3—figure supplement 4*).

Random forests classification models were applied to classify macrophage polarisation from 2P-FLIM variables when treated with FCCP (*Table 1*). To adequately train the random forests model, we removed the $\alpha_1$ 2P-FLIM variable as it exhibits a negative correlation with the $\alpha_2$ variable

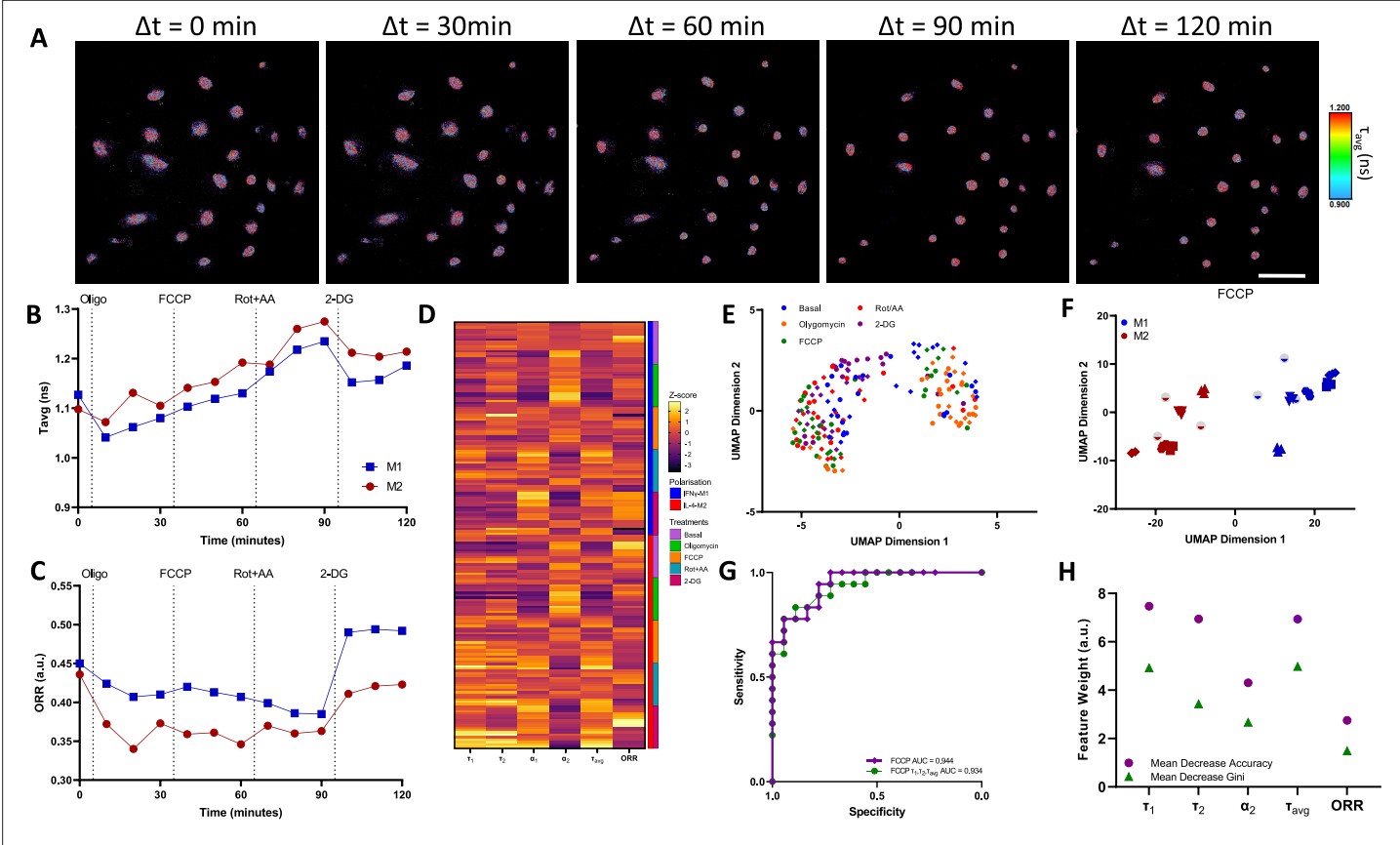

**Figure 3.** Two-photon fluorescence lifetime imaging microscopy (2P-FLIM) metabolimaging analysis. (**A**) Time-course imaging of representative (same field of view throughout) IFNγ-M1 macrophages, scale bar: 100 μm. (**B, C**) Average fluorescence lifetime ($\tau_{avg}$) and optical redox ratio (ORR) values for IFNγ-M1 and IL-4-M2 when treated sequentially with oligomycin, carbonyl cyanide-p-trifluoromethoxyphenylhydrazone (FCCP), rotenone + antimycin A and 2-deoxy-ᴅ-glucose (2-DG) of a representative donor. (**D**) z-score heatmap of 2P-FLIM acquired data for six donors separated by macrophage polarisation and metabolic inhibitor, each individual row corresponds to an imaging field. (**E**) Uniform Manifold Approximate and Projection (UMAP) plot of 2P-FLIM variables after each treatment each dot corresponds to an individual imaging field. (**F**) UMAP plot of 2P-FLIM variables after FCCP treatment, each dot corresponds to an individual imaging field. (**G**) Receiver-operator curve and area under curve values of random forests machine learning model applied to 2P-FLIM data after FCCP treatment. (**H**) 2P-FLIM weight features determined by mean decrease accuracy and mean decrease Gini of random forests model used to classify macrophages.

The online version of this article includes the following source data and figure supplement(s) for figure 3:

**Source data 1.** Fluorescence lifetime imaging microscopy (FLIM) imaging and corresponding variables measurement for each replicate, Uniform Manifold Approximate and Projection (UMAP) analysis, and machine learning (random forests) input data and coding.

**Figure supplement 1.** Two-photon fluorescence lifetime imaging microscopy (2P-FLIM) of IFNγ-M1 macrophages for basal conditions, oligomycin, carbonyl cyanide-p-trifluoromethoxyphenylhydrazone (FCCP), rotenone with antimycin A, and 2-deoxy-ᴅ-glucose treatments with the concentrations detailed in the article.

**Figure supplement 2.** Two-photon fluorescence lifetime imaging microscopy (2P-FLIM) of IL-4-M2 macrophages for basal conditions, oligomycin, carbonyl cyanide-p-trifluoromethoxyphenylhydrazone (FCCP), rotenone with antimycin A, and 2-deoxy-ᴅ-glucose treatments with the concentrations detailed in the article.

**Figure supplement 3.** Phasor analysis of NADH fluorescence lifetime imaging microscopy (FLIM) variables in basal and carbonyl cyanide-p-trifluoromethoxyphenylhydrazone (FCCP) conditions.

**Figure supplement 3—source data 1.** Phasor fluorescence lifetime imaging microscopy (FLIM) analysis raw histogram data.

**Figure supplement 4.** 2P-FLIM variables correlation matrix.

**Figure supplement 5.** Correlation matrix of two-photon fluorescence lifetime imaging microscopy (2P-FLIM) NAD(P)H variables with correlation values displayed.

**Figure supplement 6.** Dispersion plot distribution of photons/pixel per fluorescence lifetime variable value.

**Figure supplement 6—source data 1.** Dispersion plot distribution of photons/pixel per fluorescence lifetime variable value raw data.

**Table 1.** Hyper-parameters, OBB, ROC-AUC, and confusion matrix of the trained random forests model.

| Donor (no. of data points) | ntree | mtry | OOB (%) | ROC-AUC | TP | FP | FN | TN |
|---|---|---|---|---|---|---|---|---|
| All donors – full FoV (36) | 100 | 2 | 16.67 | 0.944 | 16 | 2 | 4 | 14 |

ntree, number of trees; mtry, number of variables selected for the best split at each node; OBB, out-of-bag error; ROC-AUC, area under receiver operating characteristics curve; TP, true positive; FP, false positive; FN, false negative; TN, true negative.

(*Figure 3—figure supplement 5*). ROC curves of our dataset reveal high accuracy for predicting macrophage polarisation in the full FOV during FCCP (AUC = 0.944), when using 2P-FLIM variables as predictors (*Figure 3G*).

Next, the mean decrease accuracy and mean decrease Gini returned by the random forests model reveal that $\tau_1$, $\tau_2$, and $\tau_{avg}$ are the most important 2P-FLIM variables for macrophage classification and data segregation (*Figure 3H*). When using only $\tau_1$, $\tau_2$, and $\tau_{avg}$ as the 2P-FLIM predictors for random forests training, a high prediction accuracy was still achieved (ROC-AUC = 0.934) (*Figure 3G*).

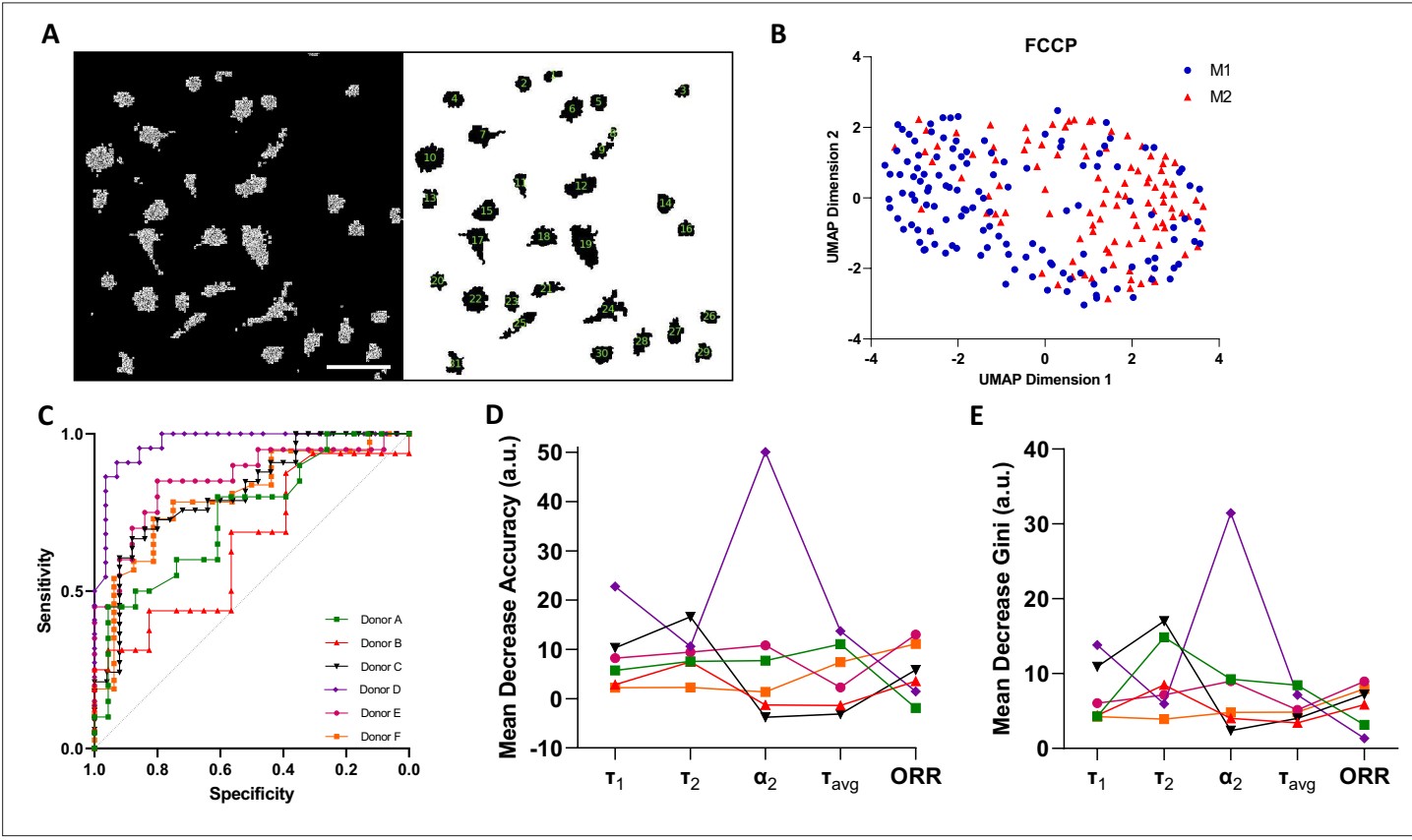

**Figure 4.** Single-cell two-photon fluorescence lifetime imaging microscopy (2P-FLIM) imaging analysis. (**A**) Single-cell analysis using a custom-built Cell Profiler script, scale bar = 100 μm. (**B**) Single-cell Uniform Manifold Approximate and Projection (UMAP) visualisation of a representative donor after carbonyl cyanide-p-trifluoromethoxyphenylhydrazone (FCCP) treatment using 2P-FLIM variables. (**C**) Receiver operating characteristics curve (ROCs) of random forests models for classification of macrophages of all human donors used in this study. (**D**) Mean decrease in accuracy and (**E**) mean decrease in Gini of each 2P-FLIM variable returned by the random forests model.

The online version of this article includes the following source data for figure 4:

**Source data 1.** Image and single-cell segmentation cell profiler coding, single-cell fluorescence lifetime imaging microscopy (FLIM) variables measurement for each replicate, and machine learning input data and coding.

**Table 2.** Hyper-parameters, OBB, ROC-AUC, and confusion matrix of donor-specific random forests models.

| Donor (no. of data points) | ntree | mtry | OOB (%) | ROC-AUC | TP | FP | FN | TN |
|---|---|---|---|---|---|---|---|---|
| A (170) | 100 | 4 | 24.41 | 0.740 | 39 | 18 | 13 | 57 |
| B (155) | 150 | 2 | 38.79 | 0.650 | 28 | 27 | 18 | 43 |
| C (232) | 250 | 3 | 32.18 | 0.813 | 67 | 29 | 27 | 51 |
| D (199) | 400 | 4 | 10.07 | 0.968 | 76 | 5 | 10 | 58 |
| E (179) | 250 | 1 | 19.40 | 0.854 | 58 | 9 | 17 | 50 |
| F (212) | 250 | 1 | 26.42 | 0.801 | 101 | 2 | 40 | 16 |

ntree, number of trees; mtry, number of variables selected for the best split at each node; OBB, out-of-bag error; ROC-AUC, area under receiver operating characteristics curve; TP, true positive; FP, false positive; FN, false negative; TN, true negative.

## 2P-FLIM classification models are sensitive to cell heterogeneity

Macrophage polarisation heterogeneity at a single-cell level was evaluated within each donor in response to the FCCP treatment (*Figure 4*). Here, we utilised Cell Profiler to evaluate and track single-cell metabolic shifts (*Figure 4A*). A representative donor UMAP implies two clusters, one majorly occupied by IFNγ-M1 macrophages and the other occupied by IL-4-M2 macrophages (*Figure 4B*).

Afterwards, we trained a random forests model for each donor. The new random forests models were trained using $\tau_{avg}$, $\tau_1$, $\tau_2$, $\alpha_2$, and ORR as predictors, the measurements of which were obtained from single cells. The values of the hyper-parameters were decided according to the OBB estimate (*Table 2*). Subsequently, we evaluated the performance of the trained random forests models by plotting ROC-AUC curves (*Figure 4C*, *Table 2*). From *Figure 4C* and *Table 2*, it is noticeable that single-cell classification performance is affected by donor variability during the FCCP treatment. Donors D and E have the highest ROC-AUC values and lowest OOB errors. Finally, for the trained random forests models, we plotted the relative importance of each 2P-FLIM variable, and found that $\tau_1$, $\tau_2$, $\tau_{avg}$, and $\alpha_2$ are the most important variables for classifying macrophage type at a single-cell level (*Figure 4D and E*).

## Discussion

In this study, we use macrophage polarisation as a model system to demonstrate the feasibility and effectiveness of the random forests model for classification, applied to 2P-FLIM parameters influenced by metabolic perturbations (*Figure 1*). The polarisation of human macrophages is often crudely described as two opposite phenotypes: classical activation (IFNγ-M1-macrophages) and alternative activation (IL-4-M2 macrophages). The higher production of TNFα in IFNγ-treated human macrophages and low IL-10 production are evidence of a macrophage classical activation (*Tokunaga et al., 2018*; *Figure 2A and B*). In contrast, treating human macrophages with IL-4, an increase in IL-10 production and higher expression of MRC1 and CCL13, are characteristic of an alternative activation of macrophages (*Martinez et al., 2006*; *Artyomov et al., 2016*; *Figure 2D–F*). Furthermore, we performed flow cytometry of polarised macrophages using antibodies to detect CD80, CD86, CD163, and CD206 surface markers, further validating the intended macrophage polarisation (*Figure 2—figure supplement 1*). Extending from this, extracellular flux analysis was performed. Extracellular flux measurements revealed a higher acidification ratio (ECAR) and a lower OCR for IFNγ-M1 macrophages during the different stages of metabolic inhibition when compared with IL-4-M2 macrophages (*Figure 2G and J*). By calculating the basal glycolysis rate and maximum glycolysis, IFNγ-M1 macrophages presented higher glycolytic rates (*Figure 2H and I*). The acidification (from ECAR) is linked with the production of lactate as a by-product of glycolysis, which reduces extracellular pH (*Wang et al., 2018*). Regarding IL-4-M2 macrophages, a reduced ECAR and increased OCR during the extracellular flux treatments (*Figure 2G and J*), together with a higher max respiration potential, were observed when compared with untreated macrophages. However, no difference was observed at the

basal respiration measures (*Figure 2K and L*). ECAR and OCR calculated results are associated with a higher dependence of OxPhos as a more active metabolic machinery in IL-4-M2 macrophages. In order to fuel the upregulation of the TCA cycle and the ETC, the mitochondria need to consume more oxygen at the complex IV site of the electron transport chain (*Van den Bossche et al., 2015*; *O'Neill et al., 2016*).

We next sought to underline 2P-FLIM as a complimentary and, more advantageously, a non-invasive spatial evaluation of macrophage metabolism reflecting macrophage-induced polarisation. Recapitulating the sequence of metabolic enzyme inhibition which formed a basis of the extracellular flux analysis, sequential 2P-FLIM micrographs of IFNγ- or IL-4-polarised macrophages were acquired (*Figure 3A*). A reduced $\tau_{avg}$, observed with IFNγ-M1 macrophages, reflects a higher relative amount of free NAD(P)H (which has characteristic short fluorescence lifetimes), indicative of glycolysis (*Walsh et al., 2013*; *Perottoni et al., 2021*). In contrast, IL-4-M2 macrophages had higher $\tau_{avg}$-reflective of OxPhos due to increased proportion of longer lifetime, protein-bound NAD(P)H (*Okkelman et al., 2019*, *Walsh et al., 2020*; *Figure 3B*). Varying interpretations of ORR are reported in previous studies (*Varone et al., 2014*; *Walsh et al., 2020*), and for this study, we adopt a lower ORR reflecting a higher fraction of NAD(P)H and a lower $FAD^+$ associated with upregulation of OxPhos (*Equation 3*; *Neto et al., 2020*). IL-4-M2 macrophages exhibited a lower ORR across the treatments in contrast with IFNγ-M1 macrophages, validating a higher dependency of IL-4-M2 macrophages on OxPhos when compared with IFNγ-M1 macrophages. A heatmap overviewing the 2P-FLIM output variables was compiled, showcasing the shifts promoted by the different treatments in both phenotype-directed human macrophages (*Figure 3D*). With IL-4-M2 macrophages, there is a noticeable increase in $\tau_1$, $\tau_2$, and $\tau_{avg}$ and an appreciable decrease in ORR when compared with IFNγ-M1-treated macrophages. The trending increase of NAD(P)H fluorescence lifetimes is further exacerbated after treatment with FCCP in IL-4-M2 macrophages. Indeed, given the ability of FLIM to imaging and measure NAD(P)H and $FAD^+$, others have employed small molecules to gain further information about the dynamics of metabolic machinery in states of disease and differentiation. For instance, during stem cell osteogenic differentiation, Guo et al. tested oligomycin A (mitochondrial respiration inhibitor) as an experimental treatment in parallel to standard osteogenic media. A decrease in NAD(P)H average lifetime was calculated reflective of reduced oxidative phosphorylation. In addition, oligomycin A treatment promoted an increase in lactate production, lower oxygen consumption, and lower osteogenic differentiation (*Guo et al., 2015*). The heterogeneous response towards metabolic inhibitors by IFNγ-M1 or IL-4-M2 macrophages yields clustering patterns in the two-dimensional projected space via the UMAP method (*Figure 3E*). When analysing each treatment individually, measurements emanating from the FCCP treatment yielded the highest segregation across all donors between IFNγ-M1 and IL-4-M2 macrophages (*Figure 3F*). In addition, our phasor analysis (*Figure 3—figure supplement 3*) also demonstrates a higher segregation of IFNγ-M1 and IL-4-M2 macrophages when using FFCP treatment. Albeit, the limitation in pre-defining values for the fluorescence lifetime variables such as $\tau_1$ and $\tau_2$ can mask important predictors for the classification problem.

Higher NAD(P)H fluorescence lifetimes such as $\tau_1$, $\tau_2$, and $\tau_{avg}$ are attributable to two major factors: increases in NADPH concentrations and microenvironmental shifts (*Blacker et al., 2014*; *Schaefer et al., 2019*). FCCP functions as an uncoupler of mitochondria inner membrane allowing unhinged proton flux to the mitochondria matrix. Consequently, this proton flux causes a reduction of mitochondrial pH, increasing effectively the fluorescence lifetime of NAD(P)H. The unhinged proton flux due to FCCP compliments existing studies whereby Schaefer et al. reported an increased NAD(P)H fluorescence lifetime due to reduced mitochondria pH elicited by FCCP treatment (*Blinova et al., 2005*; *Schaefer et al., 2017*). Another consequence of FCCP treatment is an increase in ETC activity indicated by increased oxygen consumption of IL-4-M2 macrophages (*Figure 2J*). Increased ETC activity promotes an increase of NADH and $FAD^+$ directly impacting the ORR. One would have expected the ORR to increase after the FCCP treatment as observed in IFNγ macrophages. However, for IL-4-M2 macrophages, the ORR begins to decrease. FCCP induction of maximal ETC activity increases the demand for NADH and $FADH_2$ causing a concomitant increase of the FAO and TCA cycle activity already upregulated in IL-4-M2 macrophages (*Ludtmann et al., 2014*). This demand results in a reduced pool of $FAD^+$ and an increase of NADH, effectively reducing ORR (*Ludtmann et al., 2014*; *Akie and Cooper, 2015*; *Viola et al., 2019*). The higher heterogeneity in FCCP response is due to the effect of FCCP on the mitochondrial membrane and the higher dependence of FAS/

FAO on IL-4-M2 macrophages basally (*O'Neill et al., 2016*). Blacker et al. reported the FCCP impact in great detail where, when seeking to separate NADH and NADPH fluorescence in live cells and tissues using FLIM, inhibiting mitochondrial oxidative phosphorylation in wild-type HEK293 cells using rotenone (10 µM) or uncoupling using FCCP (1 µM) (*Blacker et al., 2014*). The study of Blacker et al. provides excellent insights into NADH and NADPH dynamics and separation, and the treatment of FCCP has a similar effect on HEK293 cells as the IL-4-M2 macrophages reported in our study. Here, FCCP uncoupling promotes higher ETC, higher TCA activity impacting the ORR and, at the same time, a decreased mitochondrial pH increases the fluorescence lifetimes of NAD(P)H (*Blinova et al., 2005*; *Schaefer et al., 2017*). Regarding IFNγ-M1 macrophages, their lower dependence on ETC, TCA, FAO, and lower mitochondria membrane potential while producing most of ATP by glycolysis makes the impact of FCCP on the mitochondria and fluorescence lifetimes less pronounced.

Random forests models were applied to study the endpoint variables of the 2P-FLIM obtained during the FCCP treatment of IFNγ-M1 macrophages and IL-4-M2 macrophages. 2P-FLIM measurements contain a minimum of 24 cells per FOV and culminate in a total of 36 data points used for random forests training. For the trained random forests model, a low OBB error estimate and a high ROC-AUC value were achieved when classifying a population of IFNγ-M1 and IL-4-M2 macrophages (*Figure 3G and H*, *Table 1*). When the random forests model was trained using only three predictors, $\tau_1$, $\tau_2$, and $\tau_{avg}$, the ROC-AUC classification accuracy decreased slightly from 0.944 to 0.934 (*Figure 3G*). The feature importance ranking based on mean decrease in accuracy and mean decrease in Gini indicates that $\tau_1$, $\tau_2$, and $\tau_{avg}$ are the most important 2P-FLIM variables for classifying an IFNγ-M1 or an IL-4-M2 macrophage when subjected to the FCCP treatment (*Figure 3H*). The relative importance of the three features, $\tau_1$, $\tau_2$, and $\tau_{avg}$, implies that these FLIM features are divergent in IFNγ-M1 and IL-4-M2 macrophages. This outcome agrees with our previous results showing that FCCP highly impacts the NAD(P)H fluorescence lifetimes in IFNγ-M1 and IL-4-M2 macrophages (*Figure 3D*). The high predictive power of 2P-FLIM variables for classifying cell phenotype compliments the machine learning approach of Walsh et al., on classifying CD3[+] and CD3[+]CD8[+] T-cell activation (*Walsh et al., 2020*).

Precise regulation of macrophage activation state is key to understanding disease control, tissue homeostasis, and implant response, with this regulation shown to be directly related to macrophage intracellular metabolism (*O'Neill et al., 2016*). Therefore, impaired macrophage metabolism results in compromised homeostasis such as the case of diabetes, the foreign body response to biomaterials, obesity, or cancer (*Mantovani and Sica, 2010*; *McNelis and Olefsky, 2014*). Depending on the investigation being applied, shifts observed in cellular metabolism, cytokine production, or gene expression are typically a cumulative output from a broad population. We investigated the clustering pattern of IFNγ-M1 macrophages and IL-4-M2 macrophages using single-cell data within individual donors and found that the IFNγ-M1 macrophage and IL-4-M2 macrophage appeared as separate clusters within each donor (*Figure 4A and B*). Classifying IFNγ-M1 and IL-4-M2 macrophages at a single-cell level yielded some varied results, with four donors providing acceptable predicting performance (OOB < 31%; ROC-AUC > 0.75) (*Figure 4C and D*). There is some cell-to-cell variability which could stem from the uptake capacity of FCCP and other treatments in our experiments as well as the underlying health of the donors which is not available (*Smiley et al., 1991*; *Stiebing et al., 2017*). Nonetheless, for the four donors with the most superior performance during classification, $\tau_1$, $\tau_2$, $\tau_{avg}$, and $\alpha_2$ are the most relevant variables for the classification problem, which agrees with the case of full FoV analysis. Future efforts to improve single-cell classification include increasing the overall number of cells analysed to ensure a stronger classification. In addition, by observing phenotypic IFNγ-M1 and IL-4-M2 macrophage cell surface markers (examples include CD80, CD86, CD163, and CD206, respectively) during the imaging process by immunofluorescence or other modes of tagging, it would be possible to improve the classification efficiency (*Murray et al., 2014*).

2P-FLIM imaging has several advantages when compared with traditional metabolic assays and methods to classify and validate macrophage metabolism. 2P-FLIM enables spatial and temporal resolution in a non-invasive manner, allowing single-cell and cell-to-cell evaluations into cellular heterogeneity in a basal and interrogated mode. 2P-FLIM requires no fixation nor staining of cells and can be performed in real time with only a small number of cells. In this work, we demonstrated the feasibility of using 2P-FLIM as a tool to distinguish and classify opposing human macrophage polarisation states based on cellular metabolism and fluorescence lifetimes variables. Visualisation of the data showed a

clear clustering pattern of IFNγ-M1 and IL-4-M2 macrophages in response to FCCP during real-time imaging in a full FoV. The excellent performance of machine learning models, applied on the data extracted from the non-invasive technique, underlines further the efficiency of this workflow. This workflow can be easily adapted to non-invasively characterising macrophage polarisation in in vivo models and in vitro multicellular organoid models. These organoid models can be developed to study foreign body interactions, biomaterial assessment, pharmaceutical research and screening, and clinical applications such as disease diagnosis.

## Materials and methods

### Human blood monocyte-derived macrophage isolation

This study was approved by the research ethics committee of the School of Biochemistry and Immunology, Trinity College Dublin, and was conducted in accordance with the Declaration of Helsinki. Leucocyte-enriched buffy coats from anonymous healthy donors were obtained with permission from the Irish Blood Transfusion Board (IBTS), St. James's Hospital, Dublin. Donors provided informed written consent to the IBTS for their blood to be used for research purposes. PBMCs were isolated and differentiated into macrophages as described previously (*Mahon et al., 2020*). The purity of CD14[+]CD11b[+] macrophages was assessed by flow cytometry and was routinely >95%.

### Cytokine measurements

Macrophages ($1 \times 10^6$ cells/ml) were treated with IFNγ (20 ng/ml) or IL-4 (20 ng/ml) for 24 hr. Supernatants were harvested, and cytokine concentrations of TNFα and IL-10 were quantified by ELISA (eBioscience) according to the manufacturer's protocol.

### Real-time PCR

Macrophages ($1 \times 10^6$ cells/ml) were treated with IFNγ (20 ng/ml) or IL-4 (20 ng/ml) for 24 hr. RNA was extracted using High-Pure RNA Isolation Kits (Roche) and assessed for concentration and purity using the NanoDrop 2000c – UV-Vis spectrophotometer. RNA was equalised and reverse transcribed using the Applied Biosystems High-Capacity cDNA reverse transcription kit. Real-Time PCR Detection System (Bio-Rad Laboratories, CA) was used to detect mRNA expression of target genes. PCR reactions included iTaq Universal SYBR Green Supermix (Bio-Rad Laboratories), cDNA TaqMan fast universal PCR Master Mix and pre-designed TaqMan gene expression probes (Applied Biosystems) for CXCL9, CXCL10, MRC1, CCL13, and the housekeeping gene, 18S ribosomal RNA. The $2^{-\Delta\Delta CT}$ method was used to analyse relative gene expression.

### Seahorse analyser

Macrophages were cultured at $1 \times 10^6$ cells/ml for 6 days prior to re-seeding at $2 \times 10^5$ cells/well in a Seahorse 96-well microplate and allowed to rest for 5 hr prior to stimulation with IFNγ (20 ng/ml) and IL-4 (20 ng/ml) for 24 hr. The Seahorse cartridge plate was hydrated with XF calibrant fluid and incubated in a non-$CO_2$ incubator at 37°C for a minimum of 8 hr prior to use. Thirty minutes prior to placement into the Seahorse XF/XFe analyser, cell culture medium was replaced with complete XF assay medium (Seahorse Biosciences, supplemented with 10 mM glucose, 1 mM sodium pyruvate, 2 mM L-glutamine, and pH adjusted to 7.4) and incubated in a non-$CO_2$ incubator at 37°C. Blank wells

**Table 3.** Calculation of basal glycolysis, max glycolytic, basal respiration, and max respiration for ECAR/OCR experimental setup.

| Rate | Calculation |
|---|---|
| Basal glycolysis | Average ECAR values prior to oligomycin treatment – non-glycolytic ECAR |
| Max glycolysis | Average ECAR values after oligomycin and before FCCP treatment |
| Basal respiration | Average OCR values prior to oligomycin treatment – nonmitochondrial OCR |
| Max respiration | Average OCR values after FCCP and before rotenone/antimycin A treatment |

ECAR, extracellular acidification ratio; OCR, oxygen consumption ratio; FCCP, carbonyl cyanide-p-trifluoromethoxyphenylhydrazone.

(XF assay medium only) were prepared without cells for subtracting the background OCR and ECAR during analysis. Oligomycin (1 mM, Cayman Chemicals), FCCP (1 mM, Santa Cruz Biotechnology), Rot (500 nM), and AA (500 nM) and 2-DG (25 mM, all Sigma-Aldrich) were prepared in XF assay medium and loaded into the appropriate injection ports on the cartridge plate and incubated for 10 min in a non-$CO_2$ incubator at 37°C. OCR and ECAR were measured over time with sequential injections of oligomycin, FCCP, Rot, and AA and 2-DG. Analysis of results was performed using Wave software (Agilent Technologies). The rates of basal glycolysis, maximal glycolysis, basal respiration, and maximal respiration were calculated as detailed in the manufacturer's protocol and supplied in *Table 3*.

## Two-photon fluorescence lifetime imaging microscopy (2P-FLIM)

2P-FLIM was performed on 24 hr-polarised macrophages seeded in ibidi Luer μ-slides with a 0.8 mm channel height. 2P-FLIM was achieved using a custom upright (Olympus BX61WI) laser multiphoton microscopy system equipped with a pulsed (80 MHz) titanium: sapphire laser (Chameleon Ultra, Coherent, USA), water-immersion 25× objective (Olympus, 1.05 NA), and temperature-controlled stage at 37°C. Two-photon excitation of NAD(P)H and $FAD^+$ fluorescence was performed at the excitation wavelength of 760 and 800 nm, respectively. Several studies have reported that two-photon excitation in the range of 720–760 nm can be used to selectively excite NAD(P)H, while for $FAD^+$ a excitation wavelength above 900 nm is commonly used (*Huang et al., 2002*; *Levitt et al., 2011*). A 458/64 nm and 520/35 nm bandpass filter were used to isolate the NAD(P)H and $FAD^+$ fluorescence emissions based on their emission spectra (*Huang et al., 2002*).

512 × 512 pixel images were acquired with a pixel dwell time of 3.81 μs and 30 s collection time. A PicoHarp 300 TCSPC system operating in the time-tagged mode coupled with a photomultiplier detector assembly (PMA) hybrid detector (PicoQuanT GmbH, Germany) was used for fluorescence decay measurements, yielding 256 time bins per pixel. TCSPC requires a defined 'start,' provided by the electronics steering the laser pulse or a photodiode, and a defined 'stop' signal, realised by detection with single-photon sensitive detectors. The measuring of this time delay is repeated many times to account for the statistical variance of the fluorophore's emission. For more detailed information, the reader is referred elsewhere (*Wahl et al., 2013*).

Fluorescence lifetime images with their associated decay curves for NAD(P)H were obtained with a minimum of $1 \times 10^6$ photons peak. After imaging, the background noise was removed. This was performed by defining regions of interest (ROI) of the cells on the 2P-FLIM image. Consequently, lower values of photons/pixels are removed from analysis improving the signal-to-noise ratio (*Figure 3—figure supplement 6*).

The decay curved was generated and fitted with a double-exponential decay without including the instrument response function (IRF) (*Equation 1*).

$$I(t) = I(0) \left[ \alpha_1 e^{\frac{-t}{\tau_1}} + \alpha_2 e^{\frac{-t}{\tau_2}} \right] + C \tag{1}$$

I(t) represents the fluorescence intensity measured at time t after laser excitation; $\alpha_1$ and $\alpha_2$ represent the fraction of the overall signal proportion of a short and long component lifetime, respectively. $\tau_1$ and $\tau_2$ are the long and short lifetime components, respectively; C corresponds to background light. Chi-square statistical test was used to evaluate the goodness of multiexponential fit to the raw fluorescence decay data. In this study, all of the fluorescence lifetime fitting values with $\chi^2 < 1.3$ were considered as 'good' fits. For NAD(P)H, the double exponential decay was used to differentiate between the protein-bound ($\tau_1$) and free ($\tau_2$) NAD(P)H. The average fluorescence lifetime was calculated using *Equation 2*.

$$\tau_{avg} = \frac{(\tau_1 \times \alpha_1 + \tau_2 \times \alpha_2)}{(\alpha_1 + \alpha_2)} \tag{2}$$

Intensity-based images of NAD(P)H and $FAD^+$ were acquired, and their ratio was calculated using *Equation 3* to obtain the ORR.

$$ORR = \frac{FAD^+}{NAD(P)H} \tag{3}$$

From the images acquired using 2P-FLIM, single-cell analysis was performed using a custom-made script on Cell Profiler (*McQuin et al., 2018*). The single-cell analysis was conducted in a similar way as the global 2P-FLIM analysis.

## Macrophage classification and machine learning

UMAP was used for data visualisation and exploratory analysis of the clustering patterns in the 2P-FLIM imaging datasets for both global- and single-cell analysis (*McInnes et al., 2018*). UMAP was implemented in Python, and the plots were obtained in GraphPad. The random forests model was applied to classify IFNγ-M1 and IL-4-M2 macrophages in both full FoV and single-cell donor-specific approaches. Random forests classification was implemented in R. Random forests hyper-parameters include the number of decision trees in the forest, the number of features considered by each tree when splitting a node, and the maximal depth of each tree. The maximal depth of each tree was controlled by setting the maximal number of terminal nodes in each tree to be 8. The values of the other two hyper-parameters were determined through grid search according to the OBB error estimate (*Tables 1 and 2*). The $\alpha_1$ variable was removed from the random forests model due to its deterministic relationship with the $\alpha_2$ variable (*Figure 3—figure supplement 5*). ROCs were plotted, and the AUC values were calculated. For full FoV approach, the training dataset was used for the global analysis due to limited data size. Whereas, for the single-cell donor approach, the overall dataset was divided 75% as training datasets and 25% as testing datasets. In addition, random forests feature selection was utilised to evaluate the weight of each 2P-FLIM variable to determine its relative importance in macrophage classification for both the overall and the single-cell datasets. Support vector machine (SVM) and logistic regression models were also implemented for comparing with the random forests model and yielded similar performance with 87.5% accuracy when dividing the full FoV datapoints into 80% training and 20% testing dataset. However, neither the SVM model nor the logistic model is able to provide information on the relative importance of the predictors, and they both require an independent dataset for model validation.

## Statistics

Each experiment was performed in at least four healthy donors (defined by N) with 3–4 technical replicates run for each experiment (defined by n), depending on the assay type. Normality tests were performed to determine the normal distribution of the data. For ELISA and PCR data, one-way ANOVA and Tukey's test were used for comparing more than two groups. For Seahorse data, repeated-measures one-way ANOVA was used to account for the variance in basal metabolism across donors. All statistical analyses were performed on GraphPad Prism 9.00 (GraphPad Software).

## Acknowledgements

NN is supported by a Trinity College Dublin, Provost's PhD Award, and the TCD FLIM core unit directed by MM is supported by a SFI Infrastructure Programme: Category D Opportunistic Funds Call (16/RI/3403). This work was also partially supported by EPSRC and SFI Centre for Doctoral Training in Engineered Tissues for Discovery, Industry and Medicine, Grant Number EP/S02347X/1 and in part by a grant from Science Foundation Ireland (SFI) and the European Regional Development Fund (ERDF) under grant number 13/RC/2073_P2.

## Additional information

### Funding

| Funder | Grant reference number | Author |
| --- | --- | --- |
| Science Foundation Ireland | 16/RI/3403 | Michael G Monaghan |
| Science Foundation Ireland | EP/S02347X/1 | Michael G Monaghan |
| Science Foundation Ireland | 13/RC/2073_P2 | Michael G Monaghan<br>Aisling Dunne |

| Funder | Grant reference number | Author |
|--------|------------------------|--------|

The funders had no role in study design, data collection and interpretation, or the decision to submit the work for publication.

## Author contributions

Nuno GB Neto, Conceptualization, Data curation, Software, Formal analysis, Validation, Investigation, Visualization, Methodology, Writing – original draft, Writing – review and editing; Sinead A O'Rourke, Data curation, Formal analysis, Validation, Investigation, Methodology, Writing – review and editing; Mimi Zhang, Resources, Software, Formal analysis, Visualization, Methodology, Writing – review and editing; Hannah K Fitzgerald, Resources, Visualization; Aisling Dunne, Resources, Writing – review and editing; Michael G Monaghan, Conceptualization, Resources, Data curation, Software, Supervision, Funding acquisition, Visualization, Methodology, Writing – original draft, Project administration, Writing – review and editing

## Author ORCIDs

Nuno GB Neto ⬚ http://orcid.org/0000-0001-5467-8720
Michael G Monaghan ⬚ http://orcid.org/0000-0002-5530-4998

## Decision letter and Author response

Decision letter https://doi.org/10.7554/eLife.77373.sa1
Author response https://doi.org/10.7554/eLife.77373.sa2

# Additional files

## Supplementary files

• Transparent reporting form

## Data availability

All data generated or analysed during this study are included in the manuscript and supporting files (uploaded as source data).

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
