## [Editor Report]

The authors introduce a machine learning based classifier for M1 and M2 polarised macrophages based on autofluorescence lifetime parameters excited by two-photon excitation in the NAD(P)H emission band following during uncoupling of oxidative phosphorylation. They have identified a promising direction for use of metabolic imaging for macrophage classification.

---

## [Decision Letter]

**Decision letter after peer review:**

Thank you for submitting your article "Non-Invasive classification of macrophage polarisation by 2P-FLIM and machine learning" for consideration by *eLife*. Your article has been reviewed by 2 peer reviewers, including Michael L Dustin as the Reviewing Editor and Reviewer #1, and the evaluation has been overseen by Aleksandra Walczak as the Senior Editor. The following individual involved in the review of your submission has agreed to reveal their identity: Sergi Padilla-Parra (Reviewer #2).

Essential revisions:

1) Please, produce more data utilising the Phasor plot.

2) Produce 2D graphs clearly showing the number of photons (per pixel) vs lifetime.

3) Show all pictures of your cells with the different parameters.

4) If photons are limited leading to artifacts, increase the resolution of all your images (utilise a higher magnification/higher NA objective) and collect more photons per cell.

5) Figure 2 – Please indicate n number. Is this data from all 6 donors?

6) Figure 3 – In D, E and F, please indicate what each bar or point represents. Is this single-cell, an imaging field or200 cells? Please indicate for each panel.

7) The authors should make clear if the ROC AUC value of 0.944 is for single cells or a population? If not signal cell, how many cells are needed to reach this level?

8) The authors cite Kröger et al.'s 2021 preprint that uses lifetime parameters to classify human macrophages in vivo. Are the results here consistent with this kind of accuracy without the application of metabolic inhibitors? In this case, does in vivo environment likely serve as a discriminative condition that might separate the cells by FLIM better than the excess of oxygen in the in vitro setting?

9) The study could be strengthened further by looking at the phenotypic markers of polarisation at the single-cell level, for example, by immunofluorescence in a manner that could be correlated with the FLIM measurements. This might reveal how much the accuracy of the methods is related to some failure of macrophages to polarise in a population, rather than a true error in classification. This could be discussed as a future effort.

[Editors' note: further revisions were suggested prior to acceptance, as described below.]

Thank you for resubmitting the paper entitled "Non-Invasive classification of macrophage polarisation by 2P-FLIM and machine learning" for further consideration by *eLife*. Your revised article has been evaluated by a Senior Editor and a Reviewing Editor. We are sorry to say that we have decided that this submission will not be considered further for publication by *eLife*.

While the revisions were appreciated, they reinforced concerns about S/N. Your response related to importance of machine learning the classification led to a deeper analysis of this machine learning approach, which also suggests several weaknesses. We hope both sets of comments will be helpful in going forward with development of a robust approach to this important and interesting problem.

*Reviewer #1 (Recommendations for the authors):*

Comments on machine learning approach-

Doing large grid sweeps when cross-validating is not exactly best practice as you will be optimising for performance on the test set. Data should instead be split into train-test-val: grid sweeps should be evaluated on the test set, but final performance should be evaluated on a validation set. Otherwise, these parameters may have been overfitted to test set and may not generalise to other validation sets.

line 110 – t-SNE cannot be used for dimension reduction as it doesn’t learn a function that can be re-applied.

Given there only appear to be 6 variables used, PCA will likely be useful and faster than UMAP, and the principal components will be highly interpretable. The main utility of UMAP over PCA is that it is a non-linear transformation. PCA should still be explored to see what features make up the principal components.

139 – ROC-AUC is a bit subjective based on the number of cases: true positive, false positive, true negative and false negative should be reported too.

In figure 3: is 3F a reapplication of the UMAP learned in E, or is it a new UMAP?

Figure 4B does not at all look convincing: the M1 and M2 groups do not appear to be separated. Furthermore, was the UMAP used in this figure re-trained on this patient's data, or was it pre-trained on a different dataset? It is not clear.

There's a lot going on in figure 4: Was a new model trained for each patient, or was each patient tested on the same model? In either case, A variability of 0.937 ROC-AUC to 0.650 ROC-AUC, does not suggest that this classification model is robust.

Perhaps most importantly, It is not apparent where this score of 0.944 comes from – is this the max of cross-validation or the mean? is it on a dataset that collates all the data, or only on a subset? In particular, there is no link to the findings shown in figure 4.

This could be extended by doing ‘cross-validation’ when one of the patients is held out each time and generalisation performance is evaluated on the held-out patient.

Line 235: 'it is noticeable that single-cell classification performance is affected by donor variability during the FCCP 236 treatment' – this should be emphasised in the abstract – it is a weakness of the paper.

In Table 1 it looks like mtry and ntree are the wrong way around – having only 2-5 trees in a random forest is in no way stable, and given there are 6 features, mtry can not be more than 6 (instead of the reported range of 100 to 300).

If the researchers are doing cross-validation they should report the mean and standard deviation/S.E. for their ROC-AUC scores along with their TP/TN/FP/FN. It's not apparent if they've picked the highest ROC-AUC score they got in the cross-validations or the mean

their analysis of feature importance is fairly ad-hoc: a method like SHAP should be explored.

It appears that in the fluorescence some features are combinations of other features.

Line 486: if SVMs and logistic regressions have been done these must be reported along with their confidence intervals. Logistic regressions will also be highly interpretable. It does not appear cross-validation has been done on this data, however.

*Reviewer #2 (Recommendations for the authors):*

The authors have performed a number of analysis to try to respond to my questions. They have produced the Phasor plot for some of the data and have also presented pixel by pixel images and the photon histograms which is really valuable to understand how reliable is the data.

I have to say that after examining these data I am not convinced that the signal to noise and the limited photon collections is not affecting the results:

Figure 3 supplement 1 all figures are the same!!! There is no difference in pixel by pixel values for all conditions! The same can be seen in Figure 3 supplement 2. All figures regardless treatment conditions look the same to me. In the case of the phasor plot the differences might come from S/N as the shift in the plot is minimal and might be equivalent to your error (a few ps). In Figure 3 Supplement 4, the average lifetime clearly shows that for lower number of photons you have a shift in the lifetime which suggests that your changes in lifetime are affected by your poor signal. If you are considering the average lifetime as the mean value from a double exponential, still this shows that your calculations are affected by poor photon collection. You might consider non-fitting approach at all (for instance photon arrival time) and the number of photons would still be important. This would be better shown in a graph in which you do not bin the average lifetimes to a particular lifetime value (histograms) but instead you plot directly each photon value versus its corresponding lifetime value (dispersion plot). If you do this pixel by pixel instead of averaging your results per cell you will get a bid distribution of lifetimes that are pretty much affected by poor photon collection and you will have to determine which is the minimal amount of photons that gives a reliable lifetime.

I am sorry to say that this vision is strengthened when examining the pixel by pixel images provided with different treatments. No differences at all can be seen when taking a look at the different treatments (i.e. Olygomycin, FCCP…). Even when comparing IL-4-M2 macrophages vs IFNγ-M1 macrophages I could not see any significant difference.

Overall, I do appreciate the effort in producing all these data and I understand that there might be some differences in lifetimes that are quantified. However, the impact of the S/N and the difficulties to deconvolve background noise from real signal as shown in the histograms and also the images puts in doubt the main hypothesis of the paper.

---

## [Author Response]

Essential revisions:1) Please, produce more data utilising the Phasor plot.

We thank the reviewers for this suggestion. We have produced and added phasor plot data based on the data originally presented in the first version of this manuscript and presented the results in supplementary figure 5. We also added in the results and Discussion section an indication of this data. While we do see some separation between populations M1 and M2 when treated with FCCP and applied to a phasor analysis, the separation is quite weak. This strengthens the rationale of a machine-learning approach.

2) Produce 2D graphs clearly showing the number of photons (per pixel) vs lifetime.

We thank the reviewers for the comment and hope that we have interpreted correctly. We have calculated the number of photons per pixel of all images acquired across all fluorescence lifetime and intensity measurements. We plotted these in histograms, so the distribution of the data can be observed. For all measurements, normality tests were conducted and all data except τ_avg_ had a normal distribution. Since τ_avg_ is not a raw variable but instead derived from other fluorescence lifetime variables, we believe the double Gaussian distribution is likely due to two different populations analysed in this work (IFNγ-M1 and IL-4-M2 macrophages). In addition, while fitting these Gaussian curves we also provide their amplitude, mean, standard deviation and R^2^ value. All of these results can be found in supplementary figure 3. This comment also has overlap with comment 5 (below) where we explain the applicability of the resolution and the objective employed.

3) Show all pictures of your cells with the different parameters.

We thank the reviewers for this comment and have added all of the images for all of the parameters in supplementary figures 3 and 4. We have donor matched the extra figures with the ones present in main figure 3A. for both IFNγ-M1 and IL-4-M2 macrophages.

4) If photons are limited leading to artifacts, increase the resolution of all your images (utilise a higher magnification/higher NA objective) and collect more photons per cell.

The objective used in this study is an Olympus XLPlan 25x NA 1.05 Water Immersion objective designed specifically for 2-Photon applications. This is by far one of the most superior objectives available on the market for multiphoton excitation (Singh et al. 2015).

5) Figure 2 – Please indicate n number. Is this data from all 6 donors?

We thank the reviewers for this comment and have made corrections to make this point clearer.

To clarify- this data is from all 6 donors.

6) Figure 3 – In D, E and F, please indicate what each bar or point represents. Is this single-cell, an imaging field or200 cells? Please indicate for each panel.

We thank the reviewers for this comment and have made corrections to make this point clearer.

In Figure 3D, E and F each row and point represents an imaging field. Notably in figure 3F, each symbol is also shape-coded for each donor.

7) The authors should make clear if the ROC AUC value of 0.944 is for single cells or a population? If not signal cell, how many cells are needed to reach this level?

We thank the reviewers for this comment and have made corrections to make this point clearer. The ROC-AUC value calculated is for a population based on the image-field images obtained during the experiment. The minimum number of cells present in an image-field was 24 cells.

8) The authors cite Kröger et al.'s 2021 preprint that uses lifetime parameters to classify human macrophages in vivo. Are the results here consistent with this kind of accuracy without the application of metabolic inhibitors? In this case, does in vivo environment likely serve as a discriminative condition that might separate the cells by FLIM better than the excess of oxygen in the in vitro setting?

Yes, Kröger results have a similar accuracy to this work without resorting to metabolic inhibitors and surely, an in vivo environment can play a role on making the phenotype of M1 and M2 macrophages more distinguishable by FLIM. As described in that same work, M1 macrophages are actively engaging in phagocytosis which generates ROS leading to metabolic stress. In addition, nutrient availability, 3D environment with a complex ECM structure and the cross-talk between different cell types present in the human dermis can trigger further metabolic changes that can make it easier to evaluate macrophage polarisation using FLIM. This does evoke an age old debate- in vivo versus in vitro and absolutely the two are not directly comparable whereby even the most advance in vitro conditions (3D, bioreactors, multicellularity, ECM) do not fully capture in vivo environmental conditions, never mind those that are diseased or infected. However, our platform serves as a powerful tool for in vitro analysis and non-discriminative classification of human macrophage behaviour.

9) The study could be strengthened further by looking at the phenotypic markers of polarisation at the single-cell level, for example, by immunofluorescence in a manner that could be correlated with the FLIM measurements. This might reveal how much the accuracy of the methods is related to some failure of macrophages to polarise in a population, rather than a true error in classification. This could be discussed as a future effort.

We thank the reviewers for this comment and have added this point to the Discussion section.

We agree with this point and have included it in the discussion and a new reference which showcases fluorescent markers that could be used to determine macrophage phenotype. Indeed, in supplementary figure 6, we already use some markers to distinguish between macrophage polarizations and can serve as a starting point for improving the classification model as future work.

[Editors' note: further revisions were suggested prior to acceptance, as described below.]

While the revisions were appreciated, they reinforced concerns about S/N. Your response related to importance of machine learning the classification led to a deeper analysis of this machine learning approach, which also suggests several weaknesses. We hope both sets of comments will be helpful in going forward with development of a robust approach to this important and interesting problem.Reviewer #1 (Recommendations for the authors):Comments on machine learning approach-Doing large grid sweeps when cross-validating is not exactly best practice as you will be optimising for performance on the test set. Data should instead be split into train-test-val: grid sweeps should be evaluated on the test set, but final performance should be evaluated on a validation set. Otherwise, these parameters may have been overfitted to test set and may not generalise to other validation sets.

We thank the reviewer for this comment. Our data is comprised of small datasets with limited number of datapoints. We decided to use all of these data-points to train and validate the random forests model using cross-validation. Therefore, we do not have extra data to test our model. When we trained the random forests, we tuned the hyperparameters to achieve the smaller OBB error. The OOB allows us to estimate to the predicative error of our random forests models. In addition, we calculate the ROC curves and corresponding AUC values to evaluate the predictive ability and efficiency of the trained models. We corrected the manuscript to include this explanation.

line 110 – t-SNE cannot be used for dimension reduction as it doesn’t learn a function that can be re-applied.Given there only appear to be 6 variables used, PCA will likely be useful and faster than UMAP, and the principal components will be highly interpretable. The main utility of UMAP over PCA is that it is a non-linear transformation. PCA should still be explored to see what features make up the principal components.

Our intention is to use UMAP, PCA or t-SNE as data visualization tools only. We added a new supplemental figure 3 —figure supplement 5 where we showcase PCA and t-SNE to visualize global (ie. All treatments) and FCCP treatments FLIM data. Here, we noticed that PCA analysis enforces a clear clustering of FCCP data similar to the one observed with UMAP, whilst t-SNE does not cluster as well. We have corrected the manuscript to reflect this.

139 – ROC-AUC is a bit subjective based on the number of cases: true positive, false positive, true negative and false negative should be reported too.

Good point. To give this clarityreport these new values on table 1and 2as true positive (TP), false positive (FP), true negative (TN) and false negative (FN).

In figure 3: is 3F a reapplication of the UMAP learned in E, or is it a new UMAP?Figure 4B does not at all look convincing: the M1 and M2 groups do not appear to be separated. Furthermore, was the UMAP used in this figure re-trained on this patient's data, or was it pre-trained on a different dataset? It is not clear.

Figure 3F is a reapplication of the UMAP of figure 3E but only applied to FCCP treatment. We are using UMAP as a data visualisation technique and to classify it using random forests therefore, there is no need to enforce clear clustering pattern. The UMAP on figure 4B was re-trained on a representative donor data.

There's a lot going on in figure 4: Was a new model trained for each patient, or was each patient tested on the same model? In either case, A variability of 0.937 ROC-AUC to 0.650 ROC-AUC, does not suggest that this classification model is robust.Perhaps most importantly, It is not apparent where this score of 0.944 comes from – is this the max of cross-validation or the mean? is it on a dataset that collates all the data, or only on a subset? In particular, there is no link to the findings shown in figure 4.This could be extended by doing ‘cross-validation’ when one of the patients is held out each time and generalisation performance is evaluated on the held-out patient.

To clarify, there is a new model for each donor. Our point here is to show that he can evaluate donor heterogeneity from a single-cell point by applying random forests to single-donor data. The 0.944 value comes from a model trained using all of donor data from a large field of view treated with FCCP, specifically from figure 3F and is now reported on table 1.

Line 235: 'it is noticeable that single-cell classification performance is affected by donor variability during the FCCP 236 treatment' – this should be emphasised in the abstract – it is a weakness of the paper.

We thank the reviewer for this comment, we have corrected the manuscript and added this information on the abstract as “We uncover FLIM parameters that are pronounced under the action of carbonyl cyanide-p-trifluoromethoxyphenylhydrazone (FCCP), which strongly stratifies the phenotype of polarised human macrophages, however this performance is impacted by donor variability when analysing the data at a single-cell level”

In Table 1 it looks like mtry and ntree are the wrong way around – having only 2-5 trees in a random forest is in no way stable, and given there are 6 features, mtry can not be more than 6 (instead of the reported range of 100 to 300).

How embarrassing, the reviewer is totally correct; yes they are the wrong way around. We have corrected table 1 and table 2. Now, mtry range is from 2-5 while ntree range is 100-400.

If the researchers are doing cross-validation they should report the mean and standard deviation/S.E. for their ROC-AUC scores along with their TP/TN/FP/FN. It's not apparent if they've picked the highest ROC-AUC score they got in the cross-validations or the meantheir analysis of feature importance is fairly ad-hoc: a method like SHAP should be explored.

We thank the reviewer for this comment, we added these values to table 1. For the ROC-AUC values we did not perform cross-validation of the testing datasets and we utilised all of the data to train the random forests models. Therefore, the results obtained are based on the predictive capabilities of the trained models without a testing dataset. Using random forests allows us to evaluate the feature importance of each predicator and we have presented these results in figure 3H and figure 4E.

It appears that in the fluorescence some features are combinations of other features.

We thank the reviewer for this comment. We provide a correlation matrix in figure 4 —figure supplement 1. Here, we observed that α_2_ is negatively correlated with α_1_ and therefore is not used a predictor. The other values albeit being a mathematical obtained from other features do not show a high correlation value that would warrant removal as a variable for random forest training. For this reason, although we acquire 6 variables using 2P-FLIM, we only use 5 variables as predicators to train the random forest models.

Line 486: if SVMs and logistic regressions have been done these must be reported along with their confidence intervals. Logistic regressions will also be highly interpretable. It does not appear cross-validation has been done on this data, however.

We appreciate the reviewer’s comment. We have added the accuracy results of SVM and logistic regression directly in the manuscript. Regarding the confidence intervals, we believe they are describing the bootstrapping confidence interval. For our study, it’s not appropriate to do bootstrapping. Here, we only have 36 datapoints while performing bootstrapping requires a large dataset. In addition, bootstrapping is used more for regression and not for classification, as the assumptions underlying bootstrapping usually fail for classification problems.

Reviewer #2 (Recommendations for the authors):The authors have performed a number of analysis to try to respond to my questions. They have produced the Phasor plot for some of the data and have also presented pixel by pixel images and the photon histograms which is really valuable to understand how reliable is the data.I have to say that after examining these data I am not convinced that the signal to noise and the limited photon collections is not affecting the results:

We appreciate the reviewer’s concern. Their comments are addressed point by point below but in summary we are confident that S/N is not an issue. From the onset, before multiexponential decay fitting we set a minimum threshold of photons/pixel required for fitting. The lifetime values for bound and free NADH agree with several independent international labs and our previous measurements of pure suspended NADH. We have clarified more detail in the revised manuscript and added more supplementary figures to answer the reviewer’s queries.

Figure 3 supplement 1 all figures are the same!!! There is no difference in pixel by pixel values for all conditions! The same can be seen in Figure 3 supplement 2. All figures regardless treatment conditions look the same to me. In the case of the phasor plot the differences might come from S/N as the shift in the plot is minimal and might be equivalent to your error (a few ps).

We thank the reviewer for this comment. We have changed the colorscales of Figure 3 supplement 1 and Figure 3 supplement 2 to be more sensitive to the shifts we observed. We believe that now the impact of different metabolic treatment can be fully appreciated.

In Figure 3 Supplement 4, the average lifetime clearly shows that for lower number of photons you have a shift in the lifetime which suggests that your changes in lifetime are affected by your poor signal. If you are considering the average lifetime as the mean value from a double exponential, still this shows that your calculations are affected by poor photon collection. You might consider non-fitting approach at all (for instance photon arrival time) and the number of photons would still be important. This would be better shown in a graph in which you do not bin the average lifetimes to a particular lifetime value (histograms) but instead you plot directly each photon value versus its corresponding lifetime value (dispersion plot). If you do this pixel by pixel instead of averaging your results per cell you will get a bid distribution of lifetimes that are pretty much affected by poor photon collection and you will have to determine which is the minimal amount of photons that gives a reliable lifetime.

We thank the reviewer for this comment. We have now generated new dispersion plots to show directly each photon value (photon/pixel) to their corresponding FLIM and intensity variable. As suggested by the reviewer we are doing this in a pixel by pixel manner. We acquire our data in a 2P-FLIM system equipped with a specific objective designed for multiphoton excitation (25x, 1.05NA) and TCSPC detectors. In addition, we strive to reduce the noise of our imaging by performing it in a dark room with physical obstructions (closed system) to reduce light interference. Besides this the image is acquired during 30 seconds with an appropriate power to generate images with adequate photon counts for fitting of a multiexponential decay curve with χ^2^ values below 1.4. After imaging, we remove the background of the images obtained by using a proprietary tool of Symphotime which removes all of pixels with low photon counts in pre-determined value of photons. Referring to the new dispersion plots we do not have values below 450 photon/pixel, an observable consequence of the background removal by the user assisted by the software.

Regarding the average lifetime plot in figure 3 —figure supplement 4, we believe that the separation observed was due to the presence of two metabolically opposed cell populations. As showed in the feature importance data, τ_avg_, τ_1_ and τ_2_ are the most important parameters to distinguish both cells populations and since τavg is a mathematical combination of four fluorescence parameters, the difference of both cellular populations is estimated to be higher in the τ_avg_ plot. In addition, in the average lifetime plot, there are similar frequency values for lower and higher lifetimes. Therefore, if lower photon counts were directly impacting average fluorescence lifetimes measurements, we would observe a skewed trend towards higher frequency values revolving a particular lifetime range. Furthermore, there seems to be no trend on the other non-derived 2P-FLIM parameters.

Nonetheless, with the new dispersion plots the trend described by the reviewer is no longer present. We have randomised distributions in which we can observe that either high values of photons/pixel (2000 photons/pixel) can result in lower FLIM variables values and vice-versa. Specifically, regarding average fluorescence lifetime referred by the reviewer we no longer observe any trends. We have in our data values with 1000 photons/pixel and corresponding lifetimes of 0.95ns as well 1000 photons/pixel with 1.2ns lifetimes. With these new plots we are confident on the data being measured and on the values obtained. In the past we used these same technique and conditions

to evaluate metabolic changes that were further validated with standard biochemical techniques (DOI: 10.1172/jci.insight.139032).

I am sorry to say that this vision is strengthened when examining the pixel by pixel images provided with different treatments. No differences at all can be seen when taking a look at the different treatments (i.e. Olygomycin, FCCP…). Even when comparing IL-4-M2 macrophages vs IFNγ-M1 macrophages I could not see any significant difference.

We thank the reviewer for this comment. We have changed the limits of the colorscale of Figure 3A as well of Figure 3 —figure supplement 1 and Figure 3 —figure supplement 2. This way, we believe that the fluorescence lifetime shifts of each treatment are more pronounced and easier to be appreciated.

Overall, I do appreciate the effort in producing all these data and I understand that there might be some differences in lifetimes that are quantified. However, the impact of the S/N and the difficulties to deconvolve background noise from real signal as shown in the histograms and also the images puts in doubt the main hypothesis of the paper.

While this critique is acknowledged, we sincerely believe we have comprehensively addressed this reviewers concern.